# EV-A71 Mechanism of Entry: Receptors/Co-Receptors, Related Pathways and Inhibitors

**DOI:** 10.3390/v15030785

**Published:** 2023-03-18

**Authors:** Kanghong Hu, Rominah Onintsoa Diarimalala, Chenguang Yao, Hanluo Li, Yanhong Wei

**Affiliations:** Sino-German Biomedical Center, National “111” Center for Cellular Regulation and Molecular Pharmaceutics, Cooperative Innovation Center of Industrial Fermentation (Ministry of Education & Hubei Province), Key Laboratory of Fermentation Engineering (Ministry of Education), Hubei University of Technology, Wuhan 430068, China; hukh@hbut.edu.cn (K.H.); diarimalala1@outlook.com (R.O.D.); yaochenguang@hbut.edu.cn (C.Y.); lihanluo@hbut.edu.cn (H.L.)

**Keywords:** Enterovirus 71, receptors/co-receptors, entry, signaling pathways, inhibitors

## Abstract

Enterovirus A71, a non-enveloped single-stranded (+) RNA virus, enters host cells through three stages: attachment, endocytosis and uncoating. In recent years, receptors/co-receptors anchored on the host cell membrane and involved in this process have been continuously identified. Among these, hSCARB-2 was the first receptor revealed to specifically bind to a definite site of the EV-A71 viral capsid and plays an indispensable role during viral entry. It actually acts as the main receptor due to its ability to recognize all EV-A71 strains. In addition, PSGL-1 is the second EV-A71 receptor discovered. Unlike hSCARB-2, PSGL-1 binding is strain-specific; only 20% of EV-A71 strains isolated to date are able to recognize and bind it. Some other receptors, such as sialylated glycan, Anx 2, HS, HSP90, vimentin, nucleolin and fibronectin, were discovered successively and considered as “co-receptors” because, without hSCARB-2 or PSGL-1, they are not able to mediate entry. For cypA, prohibitin and hWARS, whether they belong to the category of receptors or of co-receptors still needs further investigation. In fact, they have shown to exhibit an hSCARB-2-independent entry. All this information has gradually enriched our knowledge of EV-A71’s early stages of infection. In addition to the availability of receptors/co-receptors for EV-A71 on host cells, the complex interaction between the virus and host proteins and various intracellular signaling pathways that are intricately connected to each other is critical for a successful EV-A71 invasion and for escaping the attack of the immune system. However, a lot remains unknown about the EV-A71 entry process. Nevertheless, researchers have been continuously interested in developing EV-A71 entry inhibitors, as this study area offers a large number of targets. To date, important progress has been made toward the development of several inhibitors targeting: receptors/co-receptors, including their soluble forms and chemically designed compounds; virus capsids, such as capsid inhibitors designed on the VP1 capsid; compounds potentially interfering with related signaling pathways, such as MAPK-, IFN- and ATR-inhibitors; and other strategies, such as siRNA and monoclonal antibodies targeting entry. The present review summarizes these latest studies, which are undoubtedly of great significance in developing a novel therapeutic approach against EV-A71.

## 1. Introduction

The *Picornaviridae* family is classified into *enteroviruses*, *rhinoviruses*, *hepatoviruses*, *cardioviruses* and *alphathoviruses* based on their morphology, physicochemical and biological properties, antigenic structures, genomic sequence and mode of replication [1]. Recently, the family of *Picornaviridae* has been updated. It is currently composed of 158 species which are grouped into 68 genera (https://www.picornaviridae.com/ (accessed on 11 December 2022)). Enterovirus type A71 (EV-A71), belonging to the genus Enterovirus, is an icosahedral, non-enveloped virus of about 30 nm in diameter. The genome, a 7.2–8.5 kb linear single-stranded (+) RNA, presents, at its 5’ end, a viral protein Vpg and an internal ribosome entry site (IRES) to initiate translation. Having only one open reading frame (ORF), the viral RNA is translated into a polyprotein divided into P1, P2 and P3 (https://viralzone.expasy.org/97 (accessed on 11 December 2022)). P1 is cleaved into VP1, VP2, VP3 and VP4, where portions of VP1, VP2 and VP3 are present at the virion surface, while VP4 is located on the internal part of the capsid. P2 and P3 are cleaved to form proteins associated with replication: 2A, 2B, 2C, 3A, 3B, 3C and 3D [2]. EV-A71 is associated mostly with hand, foot and mouth disease (HFMD), a typical disease occurring in children under the age of 5 years old. It is predominately a mild and self-limiting disease [3]. A series of HFMD outbreaks were reported in the Asia-Pacific region, in countries such as Australia [4], Cambodia [5], China [6], Japan [7], Malaysia [8], Singapore [9], South Korea [10], Taiwan [11], Thailand [12], Vietnam [13], etc., resulting in hospitalizations of children and deaths (the mortality rate ranged from <5% to 19%) [14]. Indeed, in some cases, EV-A71 causes several neurological manifestations, such as brainstem encephalitis, aseptic meningitis, acute flaccid paralysis and pulmonary complications.

The complex interaction between the virus and host proteins, plus its genome instability, makes it difficult to control and eradicate. Nevertheless, vaccines such as virus-like particles and recombinant VP1 capsids have reached clinical trials; recently, China’s National Medical Products Administration (NMPA) has licensed the commercialization of a whole EV-A71 vaccine [15]. However, to date, treatment is still at the stage of relieving symptoms only, as no FDA-approved antiviral is available. The process of invading the lipid membrane for the non-enveloped virus is not as clear as it is for the enveloped virus. Actually, three different mechanisms (depicted in detail in Figure 1A) have been reported to be involved in viral entry into cells. VP1 and VP2 viral capsids have binding sites for the main receptor, hSCARB-2 (human scavenger receptor class B, member 2), and permit EV-A71 entry into cells through clathrin-mediated endocytosis [16]. In contrast, in the absence of hSCARB-2, the other receptors/co-receptors—PSGL-1 (P-selectin glycoprotein ligand 1), sialylated glycans, Anx 2 (annexin 2), HS (heparan sulfate), HSP90 (heat shock protein 90), cypA (cyclophilin A), vimentin, nucleolin, fibronectin, prohibitin and hWARS (human tryptophanyl-tRNA synthetase)—attach to the virus and initiate caveolae-mediated or endophilin-A2-mediated endocytosis. A general outline of how EV-A71 enters host cells is described in Figure 1.

Virus–host interaction is essential for a successful infection as viruses are host-dependent parasites [17]. Receptors do not only permit entry but also define cell tropism. EV-A71 is known to target cells located in the gastrointestinal tract. However, white blood cells, cells in the respiratory tract and dentritic cells are identified as acting as secondary cell tropism [18,19,20]. This fact, as well as the presence of different symptoms of EV-A71 infection (neurologic and pulmonary complications), implies that multiple alternative entry mechanisms might exist but remain not fully elucidated. Furthermore, the existence of unidentified host proteins that interact with the virus at the stage of entry is highly possible. In fact, host receptor–viral capsid binding also triggers the activation of important signaling pathways, such as MAPK and PI3K/Akt. In addition, IFN is triggered by MDA5 recognition of the double-stranded RNA during EV-A71 replication [21]. Those mechanisms result in the activation of apoptosis and/or autophagy that permit the virus to escape the immune system and to successfully invade cells. 

Researchers have always been interested in developing anti-EV-A71 entry inhibitors. Indeed, to date, several virus receptor/co-receptor inhibitors have been developed in order to block EV-A71 at its early stages. These include: soluble forms of hSCARB-2 or PSGL-1, SP-40, lactoferrin and HS mimetics; capsid protein inhibitors that are mostly structurally designed to fit the binding sites of VP1 capsids, such as pleconaril, vapendavir and analogs; related pathway inhibitors, such as picochlorum sp.122 and durvillae Antarctica from plant extracts; and other strategies, such as siRNA and monoclonal antibodies.

In brief, this review highlights the general EV-A71 entry process by emphasizing the functions of corresponding receptors/co-receptors identified to date, the related signaling pathways and the latest progress in the development of entry inhibitors.

## 2. Enterovirus A71 Receptors/Co-Receptors

EV-A71 enters cells through receptor-mediated endocytosis, which involves three different mechanisms, i.e., (I) clathrin-mediated, (II) caveolae-mediated and (III) endophilin-A2-mediated processes. (I) In the presence of the main receptor, hSCARB2, the viral particle binds to it with or without the help of one or more co-receptors. The binding induces the recruitment of adaptor proteins on the receptor cytoplasmic tail, which afterward bind to clathrin and form “a clathrin-coated pit” (CCP), leading to EV-A71 entry through clathrin-mediated endocytosis. (II) Caveolin is a kind of protein binding to the PSGL-1 cytoplasmic tail in the presence of an actin cytoskeleton. Thus, this entrance way is called caveolae-mediated endocytosis. (III) Recently, an endophilin-A2-mediated endocytosis was identified. However, how receptors/co-receptors mediate this kind of entry has not yet been elucidated. Anyway, once endocytosis initiates, the virus is internalized and delivered to the early endosome for translocation, which is assured by the endosomal sorting complex required for transport to multivesicular bodies (ESCRT-MVBs).

hSCARB-2 was the first receptor identified and was found to act as the main receptor during EV-A71 infection by displaying its recognition activity and uncoating activity in almost all strains. As a lysosomal protein, it is neither abundant nor always present on the surface of cells [22]. Some hSCARB-2-independent entry manners have been reported successively, implying the existence of other alternative mechanisms. Indeed, PSGL-1 was the second receptor discovered. However, its binding is strain-specific, with only 20% of the EV-A71 strains isolated able to recognize it and bind to it. Since recent years, sialylated glycan, Anx 2, HS, HSP90, vimentin, nucleolin and fibronectin have been successively reported to be involved in the process of virus entry. These molecules help hSCARB-2-mediated virus adsorption, recognition and invasion. They are also known as helper receptors or co-receptors. Actually, the presence of a set of receptors/co-receptors that recognizes the capsid protein to permit attachment and entry is the first step to a successful infection. All these receptors/co-receptors are presented according to their chronologic discovery in the following sections.

### 2.1. hSCARB-2

hSCARB-2 is a receptor protein composed of 478aa, and it presents abundantly in the lysosomal membrane. Thus, it is also called the lysosomal integral membrane protein II or CD38b like-2, and it belongs to the CD36 family [23]. Initially, it was known to function in lysosomal maintenance and transportation, mainly for β-glucocerebrosidase (β-GC). It was later discovered to act as an EV-A71 cell receptor by interacting with the VP1 and (or) VP2 capsid proteins [16,24,25]. The complex structure of EV-A71-hSCARB-2 was determined via cryo-electron microscopy, and the main site of binding was located at α5 (153–163) and α7 (83–193): the G-H loop for VP1 and the E-F loop for VP2 [16]. Among the EV-A71 receptors discovered to date, hSCARB-2 is considered the main receptor as it recognizes all EV-A71 strains and induces a conformational change of the viral capsid that leads to the uncoating of the virion [25,26]. Adult mice are not susceptible to infection by EV-A71, but transgenic mice that express hSCARB-2 become susceptible to EV-A71 infection and develop similar neurological diseases to those found in humans [25,26]. In fact, EV-A71 was able to bind cells expressing hSCARB-2 on their cell surface. Moreover, the soluble form of hSCARB-2 and anti-hSCARB-2 antibodies inhibited EV-A71’s binding to it [27]. Other enteroviruses species, such as CVA7, CVA14 and CVA16, are also known to use hSCARB-2 as a receptor protein [28,29]. Thus, hSCARB-2 acts as the major receptor for EV-A71 infection and assures viral endocytosis and uncoating [30]. In addition, hSCARB-2 was associated with EV-A71 infection susceptibility by comparing the single nucleotide polymorphisms (SNPs) of the hSCARB-2 gene with the clinical HFMD severity [31]. Interestingly, the elucidation of its crystal structure permits the rational design of more and more antivirals as hSCARB-2 inhibitors.

### 2.2. PSGL-1

PSGL-1 is a glycoprotein receptor expressed on myeloid and T-lymphocyte cells. Initially, it was considered to be a counter-receptor for cell adhesion for P-, E- and L-selectin and to play an important role during inflammation [32,33]. Later, it was identified as one of the EV-A71 receptors [34]. Indeed, EV-A71 was able to bind to PSGL-1 during the infection of Jurkat T cells. The binding site was through 46, 48 and 51Tyr of PSGL-1, with amino acids located at the five-fold axis of EV-A71 [35]. However, the binding was strain-specific, and only 20% of EV-A71 strains isolated to date have been able to recognize and bind to PSGL-1. In fact, EV-A71 strains can be divided into two groups depending on their ability to bind to PSGL-1: PB (PSGL-1 binding strains) and non-PB (non-PSGL-1 binding strains), determined according to the Gly/Glu 145 of VP1, which was acting as a molecule switch [36]. In contrast to hSCARB-2, PSGL-1 may be defined as a non-main receptor and seems to be the one interacting more with the VP2 capsid protein, as mutations in the VP2 gene changed the efficiency of the infection [37]. Yen et al. found that EV-A71 receptor polymorphisms were mostly associated with the susceptibility and clinical severity of EV-A71 infection [31]. In fact, PSGL-1 is only present on the surface of immune cells such as myeloid, dentritic and lymphoid cells. However, both the PSGL-1 P- and L-selectin sites for cellular transduction signaling are recognized by the viral VP1 capsid protein. Consequently, the activation of that signaling induces viral recruitment on the surface of leucocytes and neutrophils to facilitate EV-A71’s invasion of these cells [38]. Thus, PSGL-1 acts as a key receptor for EV-A71 to reach neurologic and pulmonary cells [39]. Nevertheless, in vivo studies and pathogenicity need to be confirmed further.

### 2.3. Sialylated Glycans

Sulfated or sialylated glycans are sugar residues that are part of the glycolipids of the cell membrane. They were identified as co-receptors for several viruses [40]. Sialic acids (SA), present in terminal monosaccharides (glycan chains of glycoprotein and glycolipid), are synthesized abundantly in the gastrointestinal and respiratory epithelial systems [41]. Yang et al. investigated their potential as an EV-A71 attachment co-receptor using DLD-1 intestinal cells. As a result, the depletion of O-linked glycans as well as pretreatment with sialidase (10 mU, 2 h) and purified SA-α2, 3 Gal and SA-α2, 6Ga significantly decreased the EV-A71 infection. Thus, EV-A71 uses sialic acid-linked glycans as co-receptors [42]. 

### 2.4. Anx 2

Anx 2 is a calcium-dependent and phospholipid-binding protein belonging to the annexin family [43]. It is expressed in a wide range of cells but is abundant on the endothelium cell surface. Apart from being a co-receptor protein, Anx2 is involved in multiple functions, such as endocytosis, exocytosis, membrane domain organization, actin remodeling, signaling transduction, protein assembly, transcription, mRNA transport and DNA replication and repair [44,45]. EV-A71 was discovered using Anx2 as an attachment co-receptor. Indeed, EV-A71 binds soluble Anx2, and cells treated with antibodies against Anx2 considerably reduced the viral production. The identified binding site was between VP1 at 40aa and Anx2 at 100aa [46]. More studies should be carried out to fully understand the function of Anx 2 during EV-A71 infection, as it is known to enhance virulence, but, until now, neither viral uncoating nor entry has been reported.

### 2.5. HS

HS is a polysaccharide that exists in a wide range of mammalian cells. Structurally, it is linear and composed of a repeated arrangement of disaccharide units of N-acetylated or N-sulfated glucosamine and glucuronic acid or iduronic acid [47]. In the presence of core proteins, which are covalently attached to HS chains, it is able to recognize and bind different types of viruses [48,49]. EV-A71 was associated with HS as viral production was reduced in the presence of HS inhibitors and after pre-incubation of EV-A71 with heparin or poly-D lysine and treatment with heparinase I/II/III [50]. A previous study revealed that HS alone was unlikely to assure EV-A71 entry and infection. In fact, a successful viral entry needed the presence of hSCARB-2, as HS alone attached the viral particle through its five-fold symmetry axis in electrostatic interaction and delivered it to the main receptors [51]. 

### 2.6. HSP90

Heat shock protein 90 is a chaperone protein of 90 kDa and plays an essential role in the stabilization and/or degradation of cells under temperature stress. It is highly expressed in cells and participates in multiple signaling pathways related to cell regulation [52]. It has emerged as a promising target for cancer, and recently it has been identified to perform an important function during EV-A71 entry [53]. Indeed, HSP90 present on the cell surface was discovered to bind to the viral particle and then attach to surface receptors such as hSCARB-2 or PSGL-1 to facilitate its entry. In addition, the down-regulation of HSP90 by siRNA, antibodies or HSP90 inhibitors considerably reduced viral infection [53]. Alone, HSP90 has never been reported to mediate EV-A71 entry, but it assists by presenting it to the main receptors.

### 2.7. CypA

Cyclophilins, a family of peptidyl prolyl isomerase, are found in both prokaryotic and eukaryotic cells [54]. Cyclophilin A (cypA) was the first cyclophilin member identified, and it plays several roles: cis-trans isomerase activity, viral production (HIV, HCV), immunosuppressor, mitochondrial function, etc. [55,56,57]. The H-I loop VP1 of EV-A71 was found to interact with cypA [58]. RNAi treatment or cypA inhibitors can impair EV-A71 infection. CypA induces EV-A71 conformational changes and promotes viral entry, implying the existence of other alternative invasion routes independent of hSCARB-2. Thus, cypA represents a valuable target that is worth further study. 

### 2.8. Vimentin

Vimentin is a type III intermediate filament protein that plays multiple functions in human cells: maintaining cell shape and cytoplasm integrity, skeleton stabilization, EMT development, attachment of pathogens, etc. [59,60]. Du et al. discovered that vimentin is also one of the proteins used by EV-A71 as a cell co-receptor [61]. In fact, pull-down experiments revealed a direct interaction between vimentin and EV-A71 VP1 capsid proteins at AA151. Moreover, soluble vimentin and anti-vimentin antibodies reduced EV-A71’s binding and attachment. However, its role in virulence and entry is not yet well understood [61].

### 2.9. Nucleolin

Nucleolin is a nuclear phosphoprotein of 100-kDa that can also be found on the surface of a wide range of cells. It is involved in multiple functions: ribosome biogenesis, transcriptional regulation, chromatin remodeling, cell proliferation, apoptosis and differentiation [62]. Human immunodeficiency virus (HIV) and respiratory syncytial virus (RSV) have been reported to use nucleolin as a receptor [63,64,65]. However, EV-A71 was later identified to also interact directly with nucleolin through its VP1 capsid protein via glycoproteomics analysis. Furthermore, the knockdown of nucleolin and anti-nucleolin antibodies significantly reduced EV-A71’s binding and infection in vitro, while its expression in vivo (mouse NIH3T3) increased EV-A71’s virulence [66]. 

### 2.10. Fibronectin

Fibronectin acts as a multifunctional glycoprotein and is essential for cell adhesion, growth, migration and differentiation [67,68]. It is also able to interact with multiple molecules and mediate attachments on the cell surface. The D2 domain of fibronectin was found to interact with the amino-terminal of the EV-A71 VP1 capsid protein. It is therefore believed that fibronectin represents a cellular co-receptor for EV-A71. Indeed, EV-A71 infection is enhanced by the overexpression of fibronectin. In contrast, the treatment of infected cells with short peptides containing the Arg-Gly-Asp motif or integrin or fibronectin inhibitors considerably reduces EV-A71 infection both in vivo and in vitro [68,69]. 

### 2.11. Prohibitin

Prohibitin is a pleiotropic protein found in multiple subcellular compartments, such as the mitochondria, nuclei and plasma membrane [70]. As a multifunctional protein, it plays role in cell differentiation, proliferation, inhibition and morphogenesis [71]. Using a proteomics approach followed by mass spectrometry, Too et al. discovered that this protein is involved in EV-A71 attachment and entry [72]. Indeed, a direct interaction between prohibitin and EV-A71 was revealed by IP. In both siRNA and antibodies specific for prohibitin, Roc-A (a prohibitin inhibitor) inhibited EV-A71 in neuronal cells (NSC-34 cells) [72]. Furthermore, prohibitin might be involved in EV-A71 hSCARB-2-independent entry [73]. Thus, prohibitin is a valuable functional receptor or co-receptor that deserves further investigation.

### 2.12. hWARS

Human tryptophanyl-tRNA synthetase (hWARS) is known as an enzyme catalyzing the aminoacetylation of tRNA with tryptophan [74]. Recently, it was identified as an IFN-γ-inducible entry factor for EV-A71. In fact, the knockdown of hWARS, treatment with its competitive soluble structure, anti-hWARS antibodies or CRISPR/Cas9-mediated deletion reduced EV-A71 virulence through the attenuation of the cytopathic effect (CPE) and decrease of viral biosynthesis. The decline of EV-A71 infection was observed during both in vitro and in vivo screening (NT2 cells, L929 mouse) with or without the presence of hSCARB-2. Thus, hWARS can successfully induce viral entry in an hSCARB-2-independent manner [75].

## 3. EV-A71 Entry into Cells

Being recognized/bound by cell surface proteins is not enough for EV-A71 to successfully enter and infect cells. The viral invasion requires several processes by which the related intracellular signaling pathways cooperate, with three different mechanisms responsible for EV-A71 entry. The processes are described in detail in the following sections.

### 3.1. Adsorption to Cells

Most viruses use endocytosis, a cellular process in which extracellular substances are taken into the cytoplasm through membrane-bound vesicles such as endosomes or lysosomes. EV-A71 has been identified to use this process within multiple routes depending on the cell surface receptors/co-receptors [76]. Indeed, EV-A71 enters cells through receptor-mediated endocytosis: clathrin using a “clathrin-coated pit” (CCP) and caveolae and endophilin A2, both in the presence of dynamin and the actin cytoskeleton. The processes differ according to the type of receptor. However, once the virus is internalized, the translocation into the cytoplasm is the same, occurring via the endosomal sorting complex required for transport to multivesicular bodies (ESCRT-MVBs). Overall, three different mechanisms are involved in the process of viral entry (depicted in detail in Figure 1A).

### 3.2. hSCARB-2 Uses Clathrin-Mediated Endocytosis (CME)

As a major receptor, the crystal structure of hSCARB-2 was recently elucidated in neutral and acidic conditions [25]. Therefore, hSCARB-2 undergoes a pivotal conformational alteration at α4 and α5 at its “cap position”. Indeed, in the acidic condition, hSCARB-2 presented an opened hydrophobic tunnel that was able to bind to the EV-A71 surface protein and in turn triggered the changes of the EV-A71 structure to the “A-particle”, which uncoated and translocated through the channel [77]. The mechanism of entry was through the clathrin-mediated endocytic (CME) pathway and involved several CME key genes: AP2A1, ARRB1, CLTCL1, ARPC5, PAK1, ROCK1 and WASF1 [30,78]. The binding of the virion to hSCARB-2 triggered the first signaling and induced the binding of the adaptor proteins to the receptor cytoplasmic tail, which afterwards bound to clathrin and formed a “clathrin-coated pit (CCP)”. Furthermore, DNM1/Dynamin-1 or DNM2/Dynamin-2, a member of a class of scission proteins, pinched off the CCP and released the clathrin-coated vesicle (CCV). Auxilin and HSP70 released the clathrin basket from the vesicle and delivered its viral content to early endosomes [79]. In fact, the suppression of clathrin or dynamin via siRNA or CME chemical inhibitors reduced viral entry and infection. However, caveolin inhibitors or siRNA directed to caveolae-mediated endocytosis had no effect, suggesting that CME is the only underlying mechanism of entry through hSCARB-2 [30].

### 3.3. PSGL-1 Uses Caveolae-Mediated Endocytosis (CaME)

The mechanism of entry of EV-A71 using the PSGL-1 receptor was elucidated by inhibiting CME, which was already known to be used by the major receptor hSCARB-2 under particular pH conditions [30]. Further studies demonstrated that caveolae-mediated endocytosis is the pathway mediated by PSGL-1 [80]. Caveolae, made of caveolins (CAV1/Caveolin-1, CAV-2/Caveolin-2 and CAV3/Caveolin-3), is a specialized lipid raft that forms the flask-shaped invagination of the plasma membrane [81]. When PSGL-1 recognizes and binds to the virion, it is internalized and delivered to early endosomes for translocation. This mechanism is also pH-dependent [79,82]. 

### 3.4. Endophilin-A2-Mediated Endocytosis

Caco-2 cells are polarized human intestinal cells within the EV-A71 cell tropism. Studies of EV-A71 infection in caco-2 cells revealed that neither clathrin- nor caveolae-mediated endocytosis, as previously described, was adopted [83]. In fact, the inhibition of CME, CaME and macropynocytosis by their corresponding inhibitors or siRNA targeted genes (ENTH, EPIV 2, AP2B1 for clathrin, and CLTA, CLTB, CLTC, APFA1 for caveolin) rather enhanced than reduced EV-A71’s infection in caco-2 cells. Further studies showed that endophilin-A2-mediated endocytosis is the mechanism of entry of EV-A71 in such cells, with a dependence on dynamin-2 and the actin cytoskeleton. Indeed, while EV-A71 recognized and bound the receptors or attachments, the SH3 domain of endophilin also bound to the cytoplasmic motifs of EV-A71, activated receptors and mediated endocytosis with actin polymerization and dynamin. The translocation is assured by ESCRT-MVBs [83].

### 3.5. Relevant Signaling Pathways

Mapping host factors and signaling pathways is a very important task for the understanding of EV-A71 infection and for the development of antivirals. Currently, mitogen-activated protein kinases (MAPKs), phosphatidylinositol 3-kinase/Akt (PI3K/Akt), interferon (IFN), apoptosis and autophagy pathways have been reported to be associated with EV-A71 at the early stage of its life cycle (Figure 1B) [84,85,86]. 

#### 3.5.1. MAPK Signaling Pathway

MAPKs are a series of regulator protein kinases essential for cell proliferation, cell differentiation, innate immunity, cell movement and cell death. The MAPK pathway is one of the most studied pathways associated with EV-A71 entry and replication, as several RNA viruses induced Raf/MEK/ERK signaling for their multiplication [87]. In fact, EV-A71 activated the MEK1-ERK signaling pathway, and its suppression by siRNA or inhibitors efficiently blocked viral production. MAPK activation is initiated after the binding of viral particles to host receptors. As a result, the MAPK group, ERK, JNK and p38, is phosphorylated. Subsequently, the MEK1/ERK1 signaling cascade is activated, during which the activation of ERK1/2 by MEK1/2 is controlled by Raf. This pathway is essential for a successful EV-A71 infection, mostly in immature dendritic cells [88,89,90]. Moreover, during the study of differential gene expressions (PCR array) of MAPK signaling in EV-A71 infected RD, EV-A71 was proved to induce the activation of ERK and JNK through the release of IL-2, IL-4, IL-10 and TNF-α [91]. Furthermore, together with PI3K/Akt, the MAPK signaling pathway delayed apoptosis by downregulating GSK3 [92].

#### 3.5.2. PI3K/Akt Signaling Pathway

PI3K/Akt is an essential pathway that regulates signaling and biological processes, such as RNA processing, translation, autophagy and apoptosis. During EV-A71 infection, the PI3K pathway is favorable for virus entry. Indeed, the virus has used this pathway to escape early apoptosis in order to promote progeny propagation [93]. The underlying mechanism was similar to that for MAPK, where GSK3 is downregulated while phosphorylated PI3K/Akt activated PIP2/PIP3 and targeted the apoptosis gene in order to delay it. Moreover, the PI3K/Akt signaling pathway is directly associated with viral entry. In fact, the binding of VP1 to the major receptor or one of the co-receptors activated PI3K; in a pH-dependent manner, the virus was able to enter cells through PI3K/Akt-supported endocytosis [92]. This entry mechanism was also reported to be used by other types of viruses, such as the influenza A virus, the vaccinia virus, HCV, and the avian leucosis virus (ALV) [94,95,96,97]. 

#### 3.5.3. IFN Signaling Pathway

IFN is one of the primary responders of the immune system during viral infections. The first step, in which the virus is recognized by receptors/co-receptors on the cell surface, triggers the production of a large number of cytokines and interferons [98]. However, the virus has the ability to escape the response of the host immune system. In fact, EV-A71 proteins are able to interact with multiple host proteins responsible for the activation of the IFN signaling pathway and consequently promote viral production and escape immune responses. Interestingly, 2A^pro^ and 3C^pro^ are essential proteins of EV-A71, able to counteract and suppress type I IFN [99,100]. Moreover, EV-A71 suppresses interferon responses by blocking the JAK/signaling transducer and activator of STAT signaling by inducing karyopherin-α1 degradation [101,102]. Meanwhile, cellular endogenous microRNA-628-5p facilitates viral infection by suppressing TRAF3 signaling [98]. Moreover, type III interferon signaling restricts EV-A71 infection in goblet cells [103]. 

#### 3.5.4. Apoptosis Signaling Pathway

Apoptosis is a form of programmed cell death and a pathway used by the organism to kill infected cells. During infection, viral proteins trigger the activation of the apoptosis pathway through the activation of numerous related proteins [104]. However, EV-A71 is able to corrupt the apoptosis mechanism for its own benefit. In fact, essential molecules involved in apoptosis, such as p53, STAT1 and bax, are involved during infection. Thus, they represent an interesting target for EV-A71 drug screening [105,106]. 

#### 3.5.5. Autophagy Signaling Pathway

Together with apoptosis, autophagy plays an important function under stress situations or during an external attack on the cells such as a viral infection. It maintains homeostasis in cells and is responsible for the delivery of infected cells to lysosomes in order to eliminate them [107]. However, some viruses escape autophagy by the interaction of viral proteins with host proteins. Liu et al. compared two recombinant viruses with one attenuated VP1 and discovered that the VP1 played a vital role in disrupting autophagy through the mTOR signaling pathway [108]. 

## 4. Drug Development Targeting EV-A71 Entry

Strategies blocking viral entry include developing inhibitors targeting host receptors/co-receptors, viral capsid proteins, some key intracellular pathways and other techniques. In this field, significant progress has been made in recent years. Presented in Table 1 are several drug candidates discovered lately. These will be discussed in detail in the following section.

### 4.1. Virus Receptor Inhibitors

Soluble form of EV-A71 receptors: Several studies have demonstrated that a soluble form of the EV-A71 receptor, such as soluble hSCARB2 or PSGL-1, is able to inhibit viral attachment and entry [25,34]. It acts as a competitive inhibitor that interferes with the active site of the viral capsid so that the cell receptor can no longer bind it [109,110,111]. Moreover, glycosylation or sialylation considerably affects virus binding and subsequent entry [112]. Neuraminidase has already been used to remove cell surface sialic acids and reduce EV-A71 binding and infection rates [113].

SP40 and L-SP40: are peptides designed after the VP1 capsid protein of EV-A71. They were able to interfere with their receptors/co-receptors and inhibit viral attachment and entry. Their affinity towards multiple EV-A71 receptors was assessed by molecular docking, and nucleolin was discovered to have the highest binding affinity, followed by anx-2, hSCARB-2 and hWARS [114,115,116].

Lactoferrin: is a multifunctional iron-binding glycoprotein with a broad range of bioactivities against bacteria, fungi and viruses [117,118]. It was able to interact directly with heparin sulfate glycosaminoglycane, which served as an EV-A71 binding co-receptor. Both bovine and human lactoferrin inhibited EV-A71 uncoating and entry in vitro (IC50 = 10.5–24.5 µg/ml and 103.3–185 µg/ml, respectively) [119]. The underlying mechanism was revealed through the binding of the VP1 capsid protein [120].

Heparan sulfate mimetics: Carbohydrates such as HS and sialylated glycans (SG) are used by EV-A71 for attachment and uncoating. Several attempts to inhibit EV-A71 entry by disrupting these kinds of co-receptors have been made [112]. Heparin, heparan sulfates and pentosan polysulfate mimetics were evaluated for EV-A71 inhibition [111]. All of the three compounds exhibited antiviral activity, with heparin being the most potent (SI = 236.24). The underlying mechanism inhibiting EV-A71 infection is the hindrance of viral attachment/entry to the cell surface.

### 4.2. Virus Capsid Protein Inhibitors

Pyridyl Imidazolinone: is a capsid inhibitor known to prevent viral attachment, uncoating and the entry of several viruses such as poliovirus, the best-known picornaviridae [121,122]. Later, a series of designed pyridyl imidazolinones were tested against enterovirus 71 entry [123,124]. The mechanism was through the binding of the viral capsid protein VP1. For instance, the complex binding of WIN51711 to VP1 was reported [125]. These inhibitors represent a promising candidate for EV-A71 entry inhibition despite some resistance due to VP1 mutations [126,127]. Recently, a viral capsid inhibitor G197 and a host-targeting phosphatidylinositol 4-kinase III beta inhibitor N373, especially when used in combination, were able to significantly improve the survival of infected mice [128]. 

Pleconaril, vapendavir and their analogs: are among the best-known capsid inhibitors for picornaviruses. They were designed to be capsid-binding compounds that recognize, bind and neutralize viral protein VP1 and inhibit EV-A71 entry. They represent an excellent candidate due to their potency against almost all EV-A71 strains. Further analogs have been designed at the level of the amino group in the central chain coupled to a methylisoxazol moiety of those compounds in order to enhance their activity [129,130,131].

NF449 (4, 49, 40,409- [carbonylbis [imino-5,1,3-benzenetriyl bis(carbonyl-imino)]] tetrakis (benzene-1,3-disulfonic acid) octasodium salt): is known as a Gs-α inhibitor and has been identified to block EV-A71 activity (SI > 150). During the screening of anti-EV-A71 activity, NF449 was discovered to inhibit EV-A71 at the entry step. Furthermore, the progeny virus isolated from cells treated with NF449 presented mutations at nt 2734 and 3173 in the VPI capsid protein. Thus, NF449 is known as a capsid protein inhibitor [132]. 

Kappa carrageenan: is a sulfate polysaccharide from seaweed that exhibits broad bioactivities [133,134,135]. It has shown anti-EV-A71 activity in more than one stage of its life cycle. Kappa carrageenan inhibits EV-A71 binding, viral replication and apoptosis. It was able to directly bind to EV-A71 in a dose-dependent manner [136]. In fact, it has a similar structure as HS and is considered as its analog [137].

Sertraline: Tseng et al. screened the anti-EV-A71 activity of 774 FDA-approved drugs using a cell-based biosensor, “Hela-G2AwtR”. Among them, Sertraline, known to increase serotonin concentration by inhibiting the serotonin transporter and used traditionally as an anti-depressant, had the strongest antiviral activity against not only EV-A71 but CVA16, CVB1, CVB2, Echo9 and Echo30 as well. The selectivity index (SI) for EV-A71 was 19.92 and 19.95 for RD and Hela, respectively. Further studies revealed that its mode of action was independent of its empiric application, and the EV-A71 inhibition was at an early stage. Actually, sertraline inhibited viral entry by neutralizing endolysosomal acidification through an increase in the pH value to circa 7.0 [138]. 

Ni (2+)-chitosan beads: Chitosan is a polysaccharide known for having various biological activities and being nontoxic toward cells. It is easy to modify due to its unique structural features [139,140]. Lin et al. developed metal–ion composite chitosan beads to characterize and evaluate EV-A71 adsorption/removal. Interestingly, Ni (2+)-chitosan beads were found to be the most effective. Therefore, they represent an interesting strategy to inhibit EV-A71 infection despite the fact that the adsorption by the compound did not affect EV-A71 antigenicity [141]. 

CPZ and DNS: Since EV-A71 entry was discovered to be mainly through clathrin- and dynamin-mediated endocytosis, endocytosis inhibitors such as chlorpromazine (CPZ) and dynasore (DNS) were used to suppress EV-A71 entry and infection. However, this was shown to be not always the case, as EV-A71 used multiple entry routes and pathways [76].

PTC-209HBr: is a Bmi-1 inhibitor known as anticancer. In later studies, this small molecule was discovered to inhibit EV-A71 infection by blocking the binding between VP1 and hSCARB2 [142]. 

### 4.3. Related Signaling Pathway Inhibitors

Saururus chimensis: is an endemic plant in Asia (China, India, Japan, Korea, the Philippines and Vietnam) used to treat inflammation [143]. It has been found in later studies to exhibit different bioactivities, being anti-oxidative [144], anticancer [145,146] and antiviral [147]. Recently, this compound was detected to have anti-EV-A71 activity, targeting the MEK1/ERK signaling pathway, which is essential for EV-A71 entry and replication. In fact, treatment of Vero cells infected by EV-A71 with saururus chimensis considerably reduced the viral yield as well as the number of host proteins associated with the MEK pathway [147]. Thus, it acts as a potent EV-A71 drug candidate that also targets several essential pathways related to the virus such as AP-1 or PI3K. 

U0126: is a chemical compound that specifically inhibits MAP kinase activity by blocking MEK1/2 [147]. A study has shown that U0126 was able to block EV-A71 infection by targeting MEK/ERK signaling, a pathway related to viral entry and replication [148]. 

Formononetin: is a flavonoid present in several plants and herbs such as “the red clover”. It has multiple biological activities such as anti-inflammatory, anti-oxidant, wound healing and anti-tumor. Reportedly, it inhibited EV-A71-induced CPE by suppressing ERK, p38 and JNK through regulating cox-2/PGE2 expression, which was associated with the MAPK signaling pathway [149]. 

GS-9620: is a heterocyclic compound known as a TLR7 antagonist that exhibits antiviral activity against HBV and HIV. An in vivo study of this compound against EV-A71 showed that it inhibited viral infection by targeting the IFN and PI3K/Akt pathways [150].

Berberine: from berberis vulgaris L., is an isoquinoline alkaloid that has antiviral activity against EV-A71. The mechanism of action is through the inhibition of phosphorylated MEK/ERK-related proteins during the MAPK signaling pathway [151]. 

Se@PEI@siRNA: is a siRNA targeting VP1 and loaded with selenium nanoparticles of polyethylenimine (PEI) on the surface. It has been able to inhibit EV-A71 infection by targeting the VP1 capsid and activating bax-mediated apoptosis simultaneously. It also maintained cells in the sub-G1 phase [106]. 

Picochlorum sp.122: is a polysaccharide exhibiting antiviral activity. It reduced VP1 RNA and viral protein levels in EV-A71-infected Vero cells. The mechanism revealed that picochlorum sp.122 protected cells from apoptosis via the AKT and ATM/ATR signaling pathways [152]. 

Durvillae Antarctica: is a green alga containing polysaccharides with strong antiviral activity against multiple viruses, such as HIV, herpes simplex virus (HSV), HCV, etc. In later studies, it has been proven to reduce viral RNA replication of EV-A71 in a dose-dependent manner via inhibition of p53 and the STAT1 signaling transducer [105].

### 4.4. Other Strategies Blocking Entry 

Lipid rafts: Cholesterol plays an essential role in virus entry by endocytosis [153]. In fact, the disruption of those lipids on the cell membrane can be a target to inhibit viral entry. Zhu et al. investigated the effects of depletion or addition of cholesterol mainly by MβCD, a pharmacological agent, during EV-A71 infection. The results showed that lipid raft depletion before infection considerably inhibited EV-A71 entry and infection [154].

Monoclonal antibodies: several attempts have been made using monoclonal antibodies as candidates for EV-A71 treatment [155,156,157]. D5, H7 and C4 are monoclonal antibodies (mAbs) targeting the VP1 GH loop. Thus, studies revealed that all of them inhibited EV-A71 infection both pre- and post-infection. Further studies confirmed that those mAbs not only inhibited viral entry but also attachment, internalization, uncoating and RNA release. These mAbs were able to bind to the viral receptor binding sites of HS, hSCARB-2 and PSGL-1, thus interfering with the viral binding to these receptors [156]. Furthermore, mAb 51 and 53 belong to isotopes IgM and IgG1, respectively, and are monoclonal antibodies targeting the VP1 capsid protein. MAb 51 was able to protect 2-week-old AG129 mice from developing certain symptoms of EV-A71 infection [158]. In addition, 10D3, a neutralizing monoclonal antibody targeting a highly conserved region of the VP3 capsid, has also been identified as inhibiting EV-A71 [159].

siRNA: is a recent strategy that is being used to target EV-A71 receptors, such as hSCARB-2 or the capsid protein or related pathways, including MEK1/2, PI3K/Akt, IFN, apoptosis and autophagy [160,161]. In addition, the CME/CaME-related genes can also be targeted by siRNA in order to inhibit viral entry.

**Table 1 viruses-15-00785-t001:** A list of antivirals targeting the EV-A71 entry stage.

Drug Name	Target	EC_50_	Cell Line/Animal Model	Ref.
**Host receptor inhibitors**				
Soluble form of receptors	hSCARB-2/PSGL-1	—	RD/L-PSGL-1.1	[24,34]
SP40 and L-SP40	Nucleolin/Anx-2/hSCARB-2/hWARS/HS	6–9.3 µM	RD/Hela/HT-29/Vero/neonatal mice	[114,115,116]
Lactoferrin bovine/human	HS	10.5–24.5 µg/mL 103.3–185 µg/mL	RD	[119]
HS mimetics	HS	205 µg/mL	Vero	[111]
Heparin/heparan		290 µg/mL		
sulfates/pentosan		238 µg/mL		
**Capsid protein inhibitors**				
**Pyridyl Imidazolinone**				
❖ Compound 1	VP1	0.31–1.26 µM	Vero/MRC-5	[123,124]
❖ Compound 8		0.51–1.37 µM		
❖ Compound 20		0.04 µM		
❖ BPROZ-194		0.339 µM	RD	[126]
❖ BPROZ-299		0.04 µM		
❖ BPROZ-160		0.011 µM		
❖ BPROZ-112		0.04 µM		
❖ BPROZ-103		0.13 µM		
❖ BPROZ-101		0.0012 µM		
❖ BPROZ-074		0.008 µM		
❖ BPROZ-33		0.0088 µM		
❖ WIN51711		—		[125]
❖ G197		—		[128]
Pleconaril	VP1	0.13–0.54 µg/mL	RD/1-day-old ICR	[129]
Pirodavir		0.36–0.727 µg/mL	RD	[130]
Vapendavir		0.498–0.957 µg/mL		[131]
NF449	Capsid protein	6.5 µM		[132]
Kappa carrageenan	VP1/HS	—	Vero	[136]
Sertraline	EV-A71 entry/uncoating	1.92/1.67 µM	RD/Hela	[138]
Ni^(2+)^ chitosan beads	Capsid protein	5.5 µM	Vero	[141]
CPZ and DNS		—	RD	[76]
PTC-209HBr	hSCARB2/VP1	0.79–2.92 µM	RD/Hela/HEK293T/Vero	[142]
**Related signaling pathway inhibitors**				
Saururus Chinensis	MEK/ERK, AP-1, PI3K	—	Vero/SK--SH	[147]
U0126		3.45/3.98 µM	Mouse model	[148]
Formononetin	PI3K/Akt	—	Vero	[149]
GS-9620	MEK/ERK	73.10 µM	SK-N-SH	[150]
Berberine	bax apoptosis	—		[151]
Selenium	Akt	—		[106]
*Picochlorum* *sp.122*	P53, STAT1	13.7 µg/ml	RD	[152]
Durvillae A.		—	2-day-old Balb/c	[105]
**Other strategies blocking entry**				
Lipid rafts	Capsid protein	31.5 µg/mL	Vero	[154]
**Monoclonal antibodies**				
❖ 4E8/4C6	EV-A71 endocytosis	—	RD/Vero/Jurkat T cells/2-day-old BALB/c mice	[155]
❖ D5/H7/C4	VP1	—	Hela	[156]
❖ mAb 51/53	VP1	—	2-week-old AG129 mice	[158]
❖ 10D3	VP3	—		[159]
**siRNA**				
❖ miR-2911	VP1 GH Loop	—	Vero/Hela/HepG2/293T/L929	[160]
❖ miR-127–5p	hSCARB-2	—	293T/L929	[161]
❖ siMEK1/2	MEK/ERK	—	HEK293T	[88]

“—” means not determined in the original reference. “EC50”: the concentration applied on which 50% of total cell numbers survived representing a value of drug ability; the lower the value, the better the antiviral activity is.

## 5. Discussion and Perspectives

Since EV-A71 has provoked epidemics and severe diseases throughout the world, more and more receptors/co-receptors have been identified. However, not all of them have the ability to induce binding, uncoating and entry. In fact, EV-A71 enters cells through receptor-dependent endocytosis, and only five receptors, which are hSCARB-2, PSGL-1, cypA, prohibitin and hWARS, out of all those discovered to date, are able to mediate viral entry independently. Moreover, the presence of hSCARB-2, which is able to recognize all EV-A71 strains and even some other types of enteroviruses, such as CVA7, CVA14 and CVA16, is the only productive receptor for both attachment and uncoating, but other receptors aid in the attachment or uncoating processes [25,29]. This fact makes hSCARB-2 the main receptor and the others co-receptors. Correspondingly, the attachment mediates a conformational alteration of EV-A71 from 160S to 135S particles and permits uncoating and entry [162]. Thus, exploring the underlying mechanisms and clarifying the role of EV-A71 receptors/co-receptors are helpful to understand the scenario of viral entry and to develop well-directed inhibitors.

Although EV-A71 cell tropism is defined, its mode of entry also depends on the receptors/co-receptors available on the cell surface. Indeed, hSCARB-2 participates in clathrin-mediated endocytosis to permit EV-A71 entry, while PSGL-1 involves caveolae-mediated endocytosis. Alternatively, endophylin-A2-mediated endocytosis was observed in EV-A71-infected caco-2 cells. Previous studies confirmed the existence of other modes of entry, particularly the unknown dynamin- and hSCARB2-independent entry manner that was adopted during viral entry in A549 cells, as well as the prohibitin-, cyp-A- and hWARS-mediated endocytosis [18,75].

Several intracellular signaling pathways are involved in the viral life cycle. The MAPK signaling pathway is among the first that was found to be associated with EV-A71 entry. However, it is believed that more pathways are being controlled by EV-A71 at the early stages of the infection. Thus, mapping the entire signaling pathway will definitely provide not only new targets but more understanding of how EV-A71 infects cells and leads to serious disorders. For instance, the PI3K/Akt pathway is taken over by EV-A71 to help the virus invade. In addition, EV-A71 delays apoptosis to enhance its replication and corrupts IFN-, apoptosis- and autophagy-related signaling pathways by targeting numerous host proteins, such as STAT/KPNA1, TRAF3, type III IFN, p53 and bax. Indeed, 2A^pro^ and 3C^pro^ are essential proteins of EV-A71 that promote viral spread and, importantly, are able to interfere with IFN pathways during immune responses. 

Overall, the EV-A71 mechanism of entry is not yet fully elucidated, including the clear role of all receptors/co-receptors identified to date, the existence of unknown ones and the interaction between the host and the viral proteins that change virulence and pathogenicity. Indeed, studies demonstrated that other molecules, such as SLC35B2 and B3GAT3, are related to the successful entry of EV-A71, as their knockout but not hSCARB-2 decreased EV-A71 infection. Furthermore, the infection efficiency of EV-A71 was positively correlated with the level of host cell sulfation, not only with the amount of HS, suggesting that an unidentified sulfated protein(s) must contribute to EV-A71 infection [163]. Moreover, as this virus possesses an unstable genome, the viral capsid is prone to mutation, which leads to drug resistance and makes the development of inhibitors challenging. In addition, an appropriate animal model with similar symptoms to humans is lacking, making it difficult for an antiviral drug to reach clinical trials and FDA approval. Effectively, drug approval requires safety and efficiency tests in a large population of young children, which is hard in many cases due to ethics and culture. All these limitations need to be overcome in the near future. Last but not least, it is possible to combine all of the entry inhibitors mentioned above with other types of inhibitors that target other EV-A71 replication steps or related host proteins. For instance, 3C^pro^, an essential viral protein ensuring viral spread and infection [164,165], has been inhibited by quercetin, a flavonoid bioactive compound of plants [166]. Its combination treatment with one of the entry inhibitors will ultimately increase the drug efficiency against EV-A71 and therefore eradicate its infection and related diseases. 

## Figures and Tables

**Figure 1 viruses-15-00785-f001:**
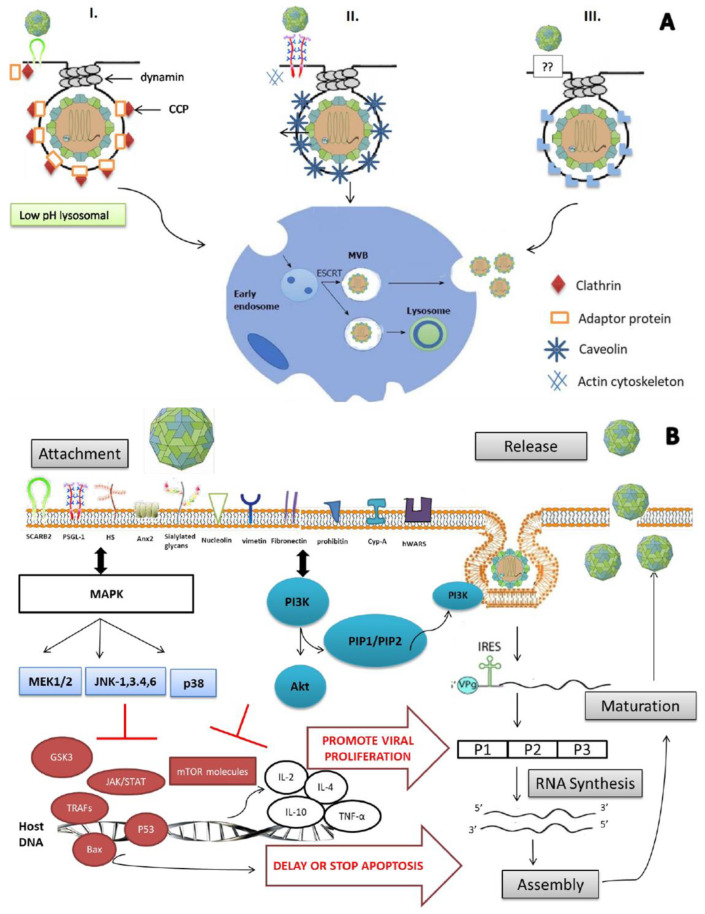
Enterovirus A71: mode of entry, virus–host interaction and viral life cycle. (**A**) Schematic illustration of the 3 different modes of entry identified to date for EV-A71 to invade cells. EV-A71 enters cells through receptor-mediated endocytosis, which includes clathrin-mediated (I), caveolae-mediated (II) and endophilin-A2-mediated (III). (I) In presence of the main receptor, hSCARB2, EV-A71 binds to it with or without the help of one or more co-receptors. The binding induces recruitment of adaptor proteins on the receptor cytoplasmic tail, which afterward bind to clathrin and form “a clathrin-coated pit” (CCP), leading to EV-A71 entry through clathrin-mediated endocytosis. hSCARB2 delivers β-GC from the ER to the lysosomes under physiological conditions. hSCARB2 is abundant in the lysosomal and endosomal compartments, and it also shuttles to the plasma membrane where it encounters EV-A71. After the binding of the virus on the cell surface, the virus–receptor complex is internalized. In the endosome or lysosome, where the pH is low, the virus initiates a conformational change that leads to uncoating. (II) In an alternative process, caveolins are the proteins binding to PSGL-1 cytoplasmic tail in the presence of actin cytoskeleton. Thus, this entrance way is called caveolae-mediated endocytosis. PSGL-1 can bind to EV-A71 and internalize via caveolin-mediated endocytosis, but PSGL-1 cannot initiate uncoating. (III) Recently, endophilin-A2-mediated endocytosis was identified. However, how receptor/co-receptor mediates this kind of entry is not yet elucidated. Once endocytosis initiates, the virus is internalized and delivered to early endosome for translocation, which is assured by the endosomal sorting complex required for transport to multivesicular bodies (ESCRT-MVBs). (**B**) EV-A71 promotes its production through interaction with intracellular signaling pathways. Once the virus is captured by main or co-receptors, several intracellular signaling pathways related to immune response are activated in order to eliminate the viral infection. However, EV-A71 has to escape immune response and overcome cellular apoptosis/autophagy for its survival by interfering with these pathways by interacting with host proteins. MAPK signaling cascade is activated after release of IL-2, IL-4, IL-10 and TNF-α. Subsequently, together with activated P13K/Akt pathway, MAPK downregulates GSK3, resulting in the delay of apoptosis. Multiple proteins involved in IFN-, apoptosis- and autophagy-related pathways are also regulated, such as JAK/STAT, TRAFs, p53, bax and mTOR. Consequently, many aspects of viral production are promoted, including viral polyprotein processing, RNA synthesis, assembly and maturation and release of new virions.

## Data Availability

Not applicable.

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
