# Peer review of "EV-A71 Mechanism of Entry: Receptors/Co-Receptors, Related Pathways and Inhibitors"

_viruses, 2023, doi:10.3390/v15030785_

Round 1

Reviewer 1 Report

Hu et al. reviewed known and recently updated knowledge of EV71 early infection and entry inhibitors. Since anti-EV-A71 is not approved for use in HFMD, it is a topic to attend to and discuss which has rarely been summarized and reported in recent years. Furthermore, the logical structure is clear and straightforward. However, there are still some issues to be resolved.

1. Since virus entries have been marked as keywords, models of viral invasion are best mentioned in the main manuscript rather than just in the figure legends.

2. In section 3.5.3 IFN signaling pathways, there is currently much understanding of how EV-A71 activates innate immune system deficiency. Furthermore, EV-A71 protein has been reported to antagonize immune activation through multiple strategies. For example, EV-A71 has identified 2A and 3C proteins that cleave critical adapters of innate immunity. The above understanding and more thoughts should be added to 3.5.3 and Discussion.

3. As the authors mentioned in the manuscript, the mechanism of EV71 entry has not been fully elucidated. More detailed unresolved issues in this area should be raised by the authors to increase the depth of thought in this review

Reviewer 2 Report

In their review, Hu et al provide an update on enterovirus A71 receptors, entry pathways, entry-related signaling pathways, as well as on inhibitors targeting these different life cycle stages. The review provides a good update on the topic but requires major improvements to meet the Viruses publication criteria. 

General comments:

The writing needs to be improved and checked by a native English speaker. Some sentences are really difficult to understand (e.g. line 53-54: "The complex host-virus interaction with the instability of the EV71 genome is the main cause that makes its eradication difficult").

Then the topics covered are very diverse and each of them could be the subject of a single review. As a result, some of these topics are treated very superficially. The authors should decide what the real purpose of the review is and expand on that topic. In particular, the section on antivirals needs to be more detailed. The title can then be changed accordingly.

The organization of the paragraphs is not always logical. Section 3.5 should be presented as a different chapter because it deals with signaling pathways, not endocytosis pathways.  Chapter 4.2 also contains inhibitors of endocytosis…

Specific comments:

The title should be corrected. First the official name of this virus is Enterovirus A71 and second "early infection" is not really appropriate nor precise enough.

The abstract should also give a brief summary on entry inhibitors.

Introduction:
The number of genus within the Picornaviridae family is outdated. Please look at https://www.picornaviridae.com/ to be precise with the current classification. In addition, the classification within the Enterovirus genus is wrong. Again the https://www.picornaviridae.com/ is the reference for picornavirus classification.

There is a precise size for the Enterovirus A71 genome

Lines 73 to 75: the main trigger of interferon response is the presence of double stranded RNA during replication. This is recognized by MDA5.

Chapter 2

There is confusion throughout the text about the role of SCARB2: this molecule is poorly expressed at the cell surface and cannot be considered as an attachment receptor. It is the uncoating receptor for most EV-A71 strains.

Line 123: what is a “count receptor” ?

It is important to note that attachment receptors such as sialic acid or heparan sulfate are not used by all EV-A71 strains. Accordingly antivirals targeting those receptors are not appropriate to treat all EV-A71 infections.

Chapter 4:
Line 401: cite exemples soluble receptors reported to inhibit EV-A71 infection

Line 447:Carrageenan (not carrageen) can also be considered as a HS analog

Line 519: “They interacted directly with HS, hSCARB-2 and PSGL-1”. This sentence is wrong and should be corrected. The mab can bind to receptor binding site on viral proteins and therefore interfere with the virus binding to these receptors. The sentence sounds more like mab could interact with the receptors.

PTC-209HBr is missing (ref: https://www.sciencedirect.com/science/article/pii/S277289272200013X)

The monoclonal antibodies Mab51 and 10D3 are also not described (ref: Lim, X.F., et al., Characterization of an isotype-dependent monoclonal antibody against linear neutralizing epitope effective for prophylaxis of enterovirus 71 infection. PLoS One, 2012. 7(1): p. e29751. And Kiener, T.K., et al., A novel universal neutralizing monoclonal antibody against enterovirus 71 that targets the highly conserved "knob" region of VP3 protein. PLoS Negl Trop Dis, 2014. 8(5): p. e2895.)

Table 1 is quite confusing. The table is informative but not easy to read and it is not obvious how the molecules relate to the references. For example, reference [87] is not linked to any description. Similarly, the GH loop of VP1 and hSCARB2 does not have an associated reference. It would be better to add separations in the table to better link the references to the antivirals. In addition some of the references listed are not appropriate. [142] doesn’t belong to monoclonal antibodies (which should be [151]), [142] belongs to Saururus Chinensis, and the ref for this is also wrong.

I would also be interesting to mention if some of these molecules have been tested in clinical trials? And currently available vaccines should figure somewhere in the review.

Reviewer 3 Report

This review attempts to integrate the latest findings on host receptors and endocytosis pathways mediating the early infection process of EV71, the signaling associated with these pathways, and the inhibitors that block these processes. However, many inaccuracies and incorrect citations throughout the manuscript undermine the paper's credibility. The section on signal transduction also lacks evidence linking it to the initial infection process, weakening the argument's overall consistency. As a result, the new perspectives and arguments presented in the review are unclear. Below is a list of examples of inaccuracies in the paper:

1.         In line 11, the term "entry" is used as the next step after "attachment" or "uncoating." However, this reviewer believes the term "endocytosis" would be more appropriate. 

2.         On line 32, there are more genera in the Picornaviridae family than are listed in the paper.

3.         On line 39, the term "IRES genome" is unclear, and 3Dpol does not encode a protein.

4.         On line 56, the text refers to "three different models," but it is unclear what these models are.

5.         On line 58, the statement "direct attachment of one of the capsid proteins (VP1, VP2 mostly)" is incorrect as the SCARB2 binding site is present in both proteins, as described in lines 109-110.

6.         On lines 69-71, the text states that "white blood cells, cells in the respiratory tract and dendritic cells are identified to act as secondary cell tropism [17]," but this is not supported by reference 17. 

7.         On lines 71-73, the statement "due to the existence of multiple alternative EV71 entry mechanisms which remain not fully elucidated" is not supported by evidence, and it is unclear why the entry mechanism is the only factor that affects tropism and pathology.

8.         On lines 91-92, there is no clear causal relationship between the diversity of tropisms and the existence of infection routes that SCARB2 does not mediate.

9.         On line 133, the reference number for Yen et al. is missing.

10.      On lines 135-136, the statement "Interestingly, EV71 infection associated with PSGL-1 receptor is the most leading to severe disease" is not supported by evidence.

11.      On lines 141-143, references 34 and 35 do not support the claim that PSGL-1 is a key receptor or that it is specific to nerve or lung cells.

12.      On line 162, the mention of "VP1-40-100aa of Anx 2" is unclear.

13.      On line 170, the use of "to form ligands" is unclear.

14.      On lines 166-176, the capsid region that binds to HS (five-fold symmetry axis), its binding mode (electrostatic interaction), and the fact that HS is an attachment receptor for cultured cell-adapted strain are not explained.

15.      On lines 251-260, reference 74 does not mention IGF1R and any claims that are not experimentally proven should not be included in the paper.

16.      On lines 267-268, the statement is incorrect because "endocytosis" refers to the process by which cells incorporate extracellular molecules into intracellular vesicles, not the process by which molecules are passed from outside to inside the cell membrane.

17.      On lines 280-282, there is no experimental evidence provided in references 26 and 77 to support the claim that the virion binding to hSCARB-2 triggers the first signaling and induces the binding of adaptor proteins to the receptor cytoplasmic tail.

18.      On line 302, Caco-2 cells are not used in reference 75.

19.      On lines 307-310, reference 82 does not provide any experimental evidence to support the claims made in this section.

Reviewer 4 Report

Review:

This review of current knowledge of the attachment and entry process of enterovirus A71 (EV-A71) is encyclopedic with a high degree of detail.  There are some corrections or necessary additions to the text that should be made:

Lines 37-40: The genome of EV-A71 is given as 7.2-8.5 kb but in GenBank I note only 7.2-7.4 kb.  Could the authors reference the 8.4 kb genome?

On lines 41-43, the authors state: “Subsequently, VP1, VP2,VP3 anchor themselves on the viral surface and function to recognize host receptors while VP4 is located on the internal part of the capsid.” This is misleading, the morphogenesis of viral capsid involves the assembly of protomers composed of VP0, VP1 and VP3. The maturation of the viral capsid containing viral RNA has the processing of VP0 into VP2 and VP4 which is internal. You could say that portions of VP1, VP2 and VP3 are present on the surface of the viral capsid but VP4 is internal.

On lines142-143: “needs further to be confirmed” should be “needs to be confirmed further”.

Line 148: were should be are.

Section 2.5 HS discusses heparin sulfate but does not include a recent paper indicating SLC35B2 regulates the tyrosine sulfation of HS and hSCAR-2 in fashion essential for EV-A71 infection (Guo D, et al. J Virol. 2022 96(9):e0204221. doi: 10.1128/jvi.02042-21). This should be included in the discussion.

Line 254: “it inhibited EV71 infection in vitro speculated through interaction of IGF1R...” should be “it inhibited EV71 in vitro which could be speculated through interaction of IGF1R...”

In Figure 1A, III, the endophilin-A2 mediated mode of entry, are the blue L shaped proteins unknown or are they endophilin-A2.  Not labeled.

In the discussion on line 533, “the only productive for successful attachment” should be “the only productive receptor for successful attachment”.  In addition from this review it appears that hSCARB-2 is the only receptor for both attachment and uncoating but other receptors or co-receptors aid in attachment or uncoating.  I suggest this sentence should be altered.

While I’m hesitant to add further references to this manuscript perhaps some already used references could be used at these sites:

On lines 43-44: Reference 3 is not a good reference for the cleavage of picornavirus proteins.  A better reference might be Palmenberg AC. Proteolytic processing of picornaviral polyprotein. Annu Rev Microbiol. 1990;44:603-23. doi: 10.1146/annurev.mi.44.100190.003131 or Jiang P, Liu Y, Ma HC, Paul AV, Wimmer E. 2014. Picornavirus morphogenesis. Microbiol Mol Biol Rev 78:418–437.

A reference for the sentence at lines 57-60 for clathrin mediated endocytosis via hSCARB-2. And on the sentence at lines 60 to 65 for other receptor modes of endocytosis.  And on the sentence at lines 73-76. As references are present in the text further in the review, perhaps the authors could just reference the section of this manuscript in which this is discussed.

Lines 90-91: A reference for the lysosomal protein hSCARB-2 being neither abundant nor present on the surface.

Lines 92-94: A reference for PSGL-1 as a receptor for EV-A71.

Lines 221-223: Neither reference 63 or 64 describe an effect of the RGD motif or inhibitors of fibronectin upon EV71 infection.  Should the reference be He et al. J Virol. 2018;92(9):e02251-17.?

Round 2

Reviewer 3 Report

The manuscript has been improved to include scientifically backed content through revisions, making it suitable for publication. However, some of the revisions have been made incorrectly, which are outlined below.

  1. In line 49, "IRES" is not an abbreviation for "internal ribosome site" but for "internal ribosome entry site."
  2. In line 53, it is a mistake to say, "VP1, VP2, and VP3 are present at the cell surface", and the correct term is "virion surface."
  3. In line 73, it is stated that "EV-A71 enters cells through direct attachment of one or two of the capsid proteins," but the authors should revise it as VP1 and VP2 have binding sites for SCARB2, as evidenced in the following article: Zhou, Daming, Yuguang Zhao, Abhay Kotecha, Elizabeth E. Fry, James T. Kelly, Xiangxi Wang, Zihe Rao, David J. Rowlands, Jingshan Ren, and David I. Stuart. 2019. "Unexpected Mode of Engagement between Enterovirus 71 and Its Receptor SCARB2." Nature Microbiology 4 (3): 414–19.
  4. In lines 292-293, "endocytosis" is a cellular process in which extracellular substances are taken into membrane-bound vesicles such as endosomes or lysosomes. As mentioned in the manuscript, the term "endocytosis, a cellular process to pass the lipid membrane barrier and enter into the cytoplasm" is incorrect. This reviewer pointed out this issue in Comment 16, but the correct revision has not been made. As this is a fundamental topic in cell biology, readers may notice that the description is incorrect, making it necessary to revise.
